# In Vitro Study of the Blood–Brain Barrier Transport of Natural Compounds Recovered from Agrifood By-Products and Microalgae

**DOI:** 10.3390/ijms24010533

**Published:** 2022-12-28

**Authors:** José David Sánchez-Martínez, Ana Rita Garcia, Gerardo Alvarez-Rivera, Alberto Valdés, Maria Alexandra Brito, Alejandro Cifuentes

**Affiliations:** 1Laboratory of Foodomics, Institute of Food Science Research, CIAL, UAM-CSIC, Nicolás Cabrera 9, 28049 Madrid, Spain; 2Research Institute for Medicines (iMed.ULisboa), Faculty of Pharmacy, Universidade de Lisboa, Av. Prof. Gama Pinto, 1649-003 Lisboa, Portugal; 3Department of Pharmaceutical Sciences and Medicines, Faculty of Pharmacy, Universidade de Lisboa, Av. Prof. Gama Pinto, 1649-003 Lisboa, Portugal

**Keywords:** neuroprotection, blood–brain barrier, green-extraction, LC/GC-q-TOF-MS, food waste, bioactive compounds

## Abstract

Agrifood by-products and microalgae represent a low-cost and valuable source of bioactive compounds with neuroprotective properties. However, the neuroprotective effectiveness of therapeutic molecules can be limited by their capacity to cross the blood–brain barrier (BBB) and reach the brain. In this research, various green extracts from *Robinia pseudoacacia* (ASFE), *Cyphomandra betacea* (T33), *Coffea arabica* (PPC1), *Olea europaea L*., (OL-SS), *Citrus sinensis* (PLE100) by-products and from the microalgae *Dunaliella salina* (DS) that have demonstrated in vitro neuroprotective potential were submitted to an in vitro BBB permeability and transport assay based on an immortalized human brain microvascular endothelial cells (HBMEC) model. Toxicity and BBB integrity tests were performed, and the transport of target bioactive molecules across the BBB were evaluated after 2 and 4 h of incubation using gas and liquid chromatography coupled to quadrupole-time-of-flight mass spectrometry (GC/LC-Q-TOF-MS). The HBMEC-BBB transport assay revealed a high permeability of representative neuroprotective compounds, such as mono- and sesquiterpenoids, phytosterols and some phenolic compounds. The obtained results from the proposed in vitro BBB cellular model provide further evidence of the neuroprotective potential of the target natural extracts, which represent a promising source of functional ingredients to be transferred into food supplements, food additives, or nutraceuticals with scientifically supported neuroprotective claims.

## 1. Introduction

In 2020, the World Health Organization placed dementia as the seventh leading cause of death worldwide. In fact, around 55 million of people were diagnosed with dementia in 2020, with forecasts reaching 78 million by 2030 [1]. Dementia has a large psychological burden on the people affected by this illness and their families, with an estimated cost of 439 billion US dollars per year in Europe [2]. Alzheimer’s disease (AD) represents nearly 60–70% of dementia cases [1]. Although the pathogenesis of AD remains misunderstood, neurofibrillary tangles (NFT) formation, amyloid-beta (Aβ) plaques formation, oxidative stress, neuroinflammation linked to lipoxygenase (LOX) overexpression, and cholinergic transmission impairment are the main hallmarks of AD, rendering the pathology multifactorial [3,4,5]. Furthermore, there is no effective cure for AD, and the currently palliative treatment consists of the use of drugs such as galantamine and rivastigmine, with the purpose of increasing the acetylcholine neurotransmitter in the synaptic clef by acetylcholinesterase (AChE) and butyrylcholinesterase (BChE) inhibition [4]. The weak effectiveness of this treatment along with some associated unpleasant side effects has compelled the search for new AD therapies [6].

For decades, several authors have turned their attention to traditional plant-based medicine. In fact, effective treatment for certain diseases has been achieved by some bioactive compounds derived from plants, such as aspirin (from willow tree bark), digoxin (from the flower, *Digitalis lanata*) and the current treatment for AD, galantamine (from *Galanthus* spp.) [7,8]. A wide variety of natural products from different origins has been tested to evaluate their neuroprotective potential in AD. Some of these bioactive compounds used in in vitro and in vivo (including humans) experiments for AD therapy in recent years include polyunsaturated fatty acids, vitamins (E, D and C), alkaloids, berberine, catechin, betaine, apigenin or curcumin. Bioactive compounds from Ginkgo biloba, *Rosmarinus officinalis L.*, saffron, green tea, colostrinin, or blueberry extracts, among others, are some promising examples in AD therapy research [9,10].

Other interesting sources of bioactive compounds are agrifood by-products, which when combined with green extraction processes, such as supercritical-fluid extraction (SFE) and pressurized liquid extraction (PLE), can help decreasing both the environmental impact of food industries and extraction processes, opening new opportunities for economical exploitation [11]. Previous studies by our group revealed that terpenoids, carotenoinds and phenolic compounds from agrifood by-products, such as unroasted cooperage woods of *Robinia pseudoacacia* (ASFE), *Cyphomandra betacea* peel (T33), *Coffea arabica* silverskin (PPC1), *Olea europaea L.* leaves (OL-SS), *Citrus sinensis* waste (PLE100) and the microalgae *Dunaliella salina* (DS) show in vitro neuroprotective potential [12,13,14,15,16]. These extracts showed a promising cholinesterase and LOX inhibition capacity, and high antioxidant activity by scavenging reactive oxygen and nitrogen species (ROS and RNS, respectively). Additionally, OL-SS, PLE100 and DS extracts have demonstrated a neuroprotective activity in the human neuroblastoma SH-SY5Y cell line against Aβ1–42 and L-glutamic acid associated toxicity [15,16,17]. However, the neuroprotective effectiveness of the drugs and bioactive compounds in the central nervous system (CNS) is limited due to the existence of physical challenges, such as the blood–brain barrier (BBB). The BBB represents a complex and dynamic barrier between the CNS and systemic circulation [18,19], and it comprises a robust interconnected network of junctional complexes between the adjacent endothelial cells (ECs), whose restrictive permeability promotes a decrease in the paracellular movement of molecules [18]. In fact, around 98% of the neurotherapeutic compounds do not cross the BBB [19], limiting the possibility of bioactive compounds entering the brain parenchyma. In this line, in silico, in vitro and in vivo models have been used to study the transport of molecules, including natural compounds, across the BBB [20,21]. In silico prediction models of BBB permeation have the advantage of allowing a rapid and inexpensive screening of lead compounds, speeding up the drug discovery process in the CNS area [22], but require experimental validation. On the other hand, in vivo models provide the most reliable information regarding the ability of compounds to access the CNS, but animal models cannot be used as high-throughput screening methods due to the high numbers of molecules studied in early drug discovery stages [23]. Among the in vitro models, the parallel artificial membrane permeability assay for the BBB (PAMPA-BBB) involves a high-throughput non-cell-based permeation test, validated to study the rate of transcellular passive diffusion through the BBB [24]. Nevertheless, the use of PAMPA methodology could lead to underestimation of the permeability of cellular transport-dependent molecules. To overcome the limitations of the lack of cellular transport, BBB cell-based models relying on confluent cultures of brain microvascular endothelial cells, considering the anatomic basis of the BBB [19], have been used. A comparison of immortalized human brain capillary endothelial cell lines revealed human brain microvascular endothelial cells (HBMEC) as the most suitable human cell line for an in vitro BBB model concerning barrier tightness [25]. Accordingly, this cell line has been widely used, particularly in BBB permeation studies in which endothelial cells are cultured on semipermeable filters (transwell inserts) with two well-defined compartments: the apical or upper compartment, which can be considered as the “blood-side” and the basolateral or lower compartment, which is considered the “brain side” [26,27,28].

The aim of the present work is to evaluate the BBB permeability of different natural compounds from agrifood by-products and microalgae extracts by using the HBMEC cell line in a transwell-based system. Preliminary studies on the potential cellular cytotoxicity and the effect on the BBB integrity (barrier tightness and paracellular permeability) were also investigated.

## 2. Results and Discussion

In this study, we utilized a widely used in vitro model of the BBB, relying on confluent monolayers of the cell line of HBMEC grown onto a semi-permeable membrane (transwell insert) defining two chambers: an upper or luminal one that mimics the blood side and a lower or basolateral one that mimics the brain side [27,28,29]. The extracts component’s ability to cross the BBB endothelium was determined following the addition of the extracts to the “blood” side and their quantification in both “blood” and “brain” sides by GC/LC-Q-TOF-MS, as depicted in Figure 1.

### 2.1. Evaluation of the Toxicity of Agrifood By-Products and Microalgae Extracts on HBMEC

Since the major objective of the present work is to investigate the transport of bioactive compounds from extracts across the brain microvascular endothelium, without damaging the BBB, it is essential to firstly evaluate the safety of the extracts used. Therefore, a viability assay by the MTT test was performed in HBMEC cultures to determine the safe concentrations of each of the six extracts (Figure 2). The tested concentrations were selected based on previous viability tests performed for the same extracts on other human cell lines, namely neuron-like (SH-SY5Y), kidney (HK-2) and monocyte THP-1) cells [12,13,14,15,16]. The use of a higher extract concentration could improve the concentration of natural compound transported to the lower compartment and their detection by GC-LC/MS equipment; however, toxic concentrations could induce alterations in HBMEC integrity and permeability. Our results demonstrated that ASFE and OL-SS extracts promote a significant increase in cells viability at most of the studied concentrations, and that DS and T33 favor the viability at a single concentration, whereas the PPC1 extract is safe for the HBMEC endothelium at all tested concentrations. In contrast, T33 and PLE100 extracts exhibited toxicity at the highest concentrations tested (60 µg/mL for T33; 30 and 60 µg/mL for PLE100). Based on these results, the maximum non-toxic concentration of each extract was selected to assess the BBB integrity and the transport assays: ASFE (24 µg/mL), PPC1 (50 µg/mL), DS (10 µg/mL) OL-SS (40 µg/mL), PLE100 (15 µg/mL) and T33 (30 µg/mL). In line with our previous results in other cell lines, PPC1, DS and OL-SS extracts were safe at all the concentrations tested. Regarding the PLE100 extract, it showed some toxicity at the highest concentration tested (60 µg/mL) in HBMEC and SH-SY5Y cell lines. On the other hand, the ASFE extract at 24 µg/mL was safe for HMBECs, but not for SH-SY5Y cells; finally, the highest concentration of T33 extract (60 µg/mL) was toxic for HBMEC, but not for SH-SY5Y and HK-2 cells [12,13,14,15,16].

### 2.2. Effect of Agrifood By-Products and Microalgae Extracts on BBB Integrity

BBB integrity experiments are crucial to determine the normal function of the in vitro BBB model and to study the specific transport capacity of each natural bioactive compound. In this regard, an increased presence of Na-F in the lower compartment indicates a permeability increase caused by alterations on the monolayer integrity and, as a consequence, the loss of barrier functionality [30]. On the other hand, the reduction in TEER values during this assay indicates the disruption of tight junctions and loss of BBB properties [19]. Based on these premises, the integrity of the BBB was determined by the measurement of Na-F flux (Figure 3a) and TEER (Figure 3b). The results showed that PPC1 was the only extract that damaged the HBMEC monolayer, as revealed by the increment of Pe values after 2 and 4 h (*p* < 0.05 and *p* < 0.01, respectively). For OL-SS, ASFE, PLE100 and T33, no significant differences were found in Pe values at 2 h, whereas at 4 h PLE100 and T33 even induced barrier tightness (*p* < 0.05 and *p* < 0.01, respectively), as indicated by the decreased permeability (Figure 3a). On the other hand, for TEER, none of the extracts affected the values not significantly lower than the control samples. T33 extract seemed to improve the TEER values, although it was not significant compared to the control. In this line, Kam et al. (2012) summarized the protective effect of several natural products against BBB breakdown [18]. Among them, terpenoids and phenolic compounds from different plant sources present preventive effects towards BBB breakdown in various animal and cellular models. The same applies to triterpenoids from *Olea europaea* or caffeic acids present in our extracts [18]. Monoterpenoid-enriched extracts from *Lavandula officinalis* also showed a protective effect against BBB permeability in an induced rat model [31]. These results are in line with our BBB integrity experiments, especially for PLE100 and T33 extracts, whose large content in terpenoids and phenolic compounds could protect and improve the BBB functionality. Moreover, the described results after PLE100 extract exposition could be also explained by the large content of phytosterols in this extract, which have shown a beneficial role in BBB endothelial cells’ integrity, including anti-apoptotic properties [32]. In fact, some authors have described the possibility of some bioactive compounds exerting neuroprotective effects without crossing the BBB, due to the direct effect in the BBB integrity (the disruption and the abnormal BBB permeability is related with some diseases, including AD) [33]. The authors proposed that the beneficial mechanism of natural compounds in the BBB integrity may be related to the modulation of neuroinflammatory signaling cascades and oxidative stress related to BBB breakdown and neurodegeneration [18,31,32,33].

The contradictory effect observed in PPC1 extract may be explained by its caffeine content, as high caffeine concentration can destabilize intracellular signaling pathways including levels of intracellular calcium and cAMP (cyclic adenosine monophosphate) [34]. In this regard, another possible explanation of the increasing permeability could be the influence of caffeine in other cellular transporters involved in the Na-F flux. Thus, the major presence of Na-F may be derived from increased Na-F transportation, which could explain the normal TEER value and the higher permeability during PPC1 extract exposition [35,36]. In this sense, although a non-toxic concentration (50 µg/mL) was used for the PPC1 extract and no alterations in TEER experiments were observed, the abnormal increment of Na-F Pe values at each time point indicate alterations in the HBMEC monolayer integrity and defects in the BBB transport selectivity. For this reason, the PPC1 extract analysis was not included in the study of transport efficiency of its bioactive compounds.

### 2.3. Evaluation of Natural Compound Transport across the BBB Endothelium

PLE100, OL-SS, ASFE and T33 extracts were submitted to an extensive GC/LC-MS analysis to determine the BBB in vitro permeation potential of their bioactive compounds (Figure 4). These extracts have been previously studied in different in vitro neuroprotective assays by our group (Table 1) and the PAMPA-BBB permeability model to explore the passive diffusion of the neuroprotective compounds [12,13,14,15,16]; however, some limitations of the PAMPA-BBB model arise, such as the lack of paracellular permeability or the absence of active cellular transport. These problems can be solved using more complex BBB models, such as HBMEC confluent monolayers platted onto transwell inserts, mentioned above. Figure 4 shows the results corresponding to the bioactive compounds transport efficiency for PLE100, OL-SS, ASFE and T33 extracts, following 2 and 4 h incubation, in the cellular model. 

Regarding the PLE100 extract, β-amyrin was not detected, indicating that it does not cross the BBB endothelium. The remaining compounds presented a time-dependent permeation. The most permeant molecules were valencene, campesterol and (−)-α-panasinsen, showing about 75% transport efficiency with a 4 h incubation. The brain accessibility of valencene reinforces that this is one of the most valuable neuroprotective sesquiterpenoids identified in the PLE100 extract [15]. Other compounds, such as elemol, γ-silenene or β-panasinsene, showed similar transport efficiency between the two timepoints, probably because their maximum transport capacity was reached after 2 h. In line with our previous study performed with a PAMPA-BBB methodology, β-amyrin was not detected after the BBB transport experiment; however, some compounds such as L-α-terpineol, nerol, (-)-aristolene and squalene (that could not cross the PAMPA-BBB system) [37] were detected in the basolateral compartment after the transport assay in the transwell-based model. For those compounds that cannot cross the BBB trough passive diffusion (PAMPA-BBB), it is expected that some cellular transporting mechanisms could be involved, as it was recently discussed by Velásquez-Jiménez et al. [38].

In the case of OL-SS extract, similar results to the PAMPA-BBB model were obtained: the majority of terpenoids showed a notable capacity to cross the BBB endothelium; however, β-amyrin, α-amyrin and lupeol acetate were not detected in the PAMPA-BBB model nor in the basolateral compartment (“brain side”) of the HBMEC-BBB model. In this extract, cymenol, (-)-globulol, 3-hydroxy-β-damascone and caryophyllene oxide showed the highest permeability capacity with transport efficiency values surpassing 75% after 4 h of incubation. Furthermore, the triterpenoid uvaol was detected in the acceptor well of the PAMPA-BBB system and in the basolateral compartment of the HBMEC cells. These observations are in line with the fact that this compound was previously correlated with the in vitro neuroprotective properties of the OL-SS extract [14]. The transport process across the BBB of terpenoids is still unclear, and the available literature concerning the BBB permeability of terpenoids is very limited. Only in a few studies has the presence of some isolated terpenoids, such as limonene, nootkatone or elemene, been detected in CNS after post mortem tissue analysis in animal models, [39,40,41] or by using in silico prediction models [42]. On the other hand, phytosterols, such as campesterol, stigmasterol, fucosterol or γ-sitosterol, have been detected in human brains, and the possible transport mechanism across the BBB has been described by Vanmierlo et al. [32]. These authors suggested that phytosterols can reach the CNS by the scavenging mediated receptor class B member 1 (SR-BI), that it is expressed at the apical membrane of BBB endothelial cells and is also involved in the sterol metabolism [32].

With regard to the T33 extract, all the detected compounds in the basolateral compartment were also detected in the “brain side” (except rutin) (Figure 4). Interestingly, some phenolic compounds present in this extract, such as sinapic acid, O-acetyl-quinic acid, caffeoyl hexoside, syringic acid hexoside, hexosyl methyl ferulate, methyl feruloyl quinate and isoquercitrin, were previously determined as compounds that were not capable of crossing the in vitro PAMPA-BBB [12], and, therefore, some active cellular transport could be responsible for their transport across the HBMEC monolayer.

Similar to the T33 extract, most of the phenolic compounds of the ASFE extract showed good transport efficiency. Only a few compounds, such as hesperidin, coutaric acid and kaempferol diacetylglucoside rhamnoside, were not detected in the lower compartment after the transport assay through the HBMEC confluent monolayers. Again, there are some discrepancies with previous PAMPA-BBB results [12] as some phenolic compounds, such as pyrogallol, ferulic acid, sinapic acid, aromadendrin, ellagic acid, caftaric acid, ε-viniferin, ganolucidic acid B and procyanidin B2, could reach the “brain side” of the HBMEC model, but were not detected in the lower well of the PAMPA-BBB model. Some pyrogallol derivatives were also found in the brain compartment in a previous study with the same HBMEC-BBB transwell model [27].

**Table 1 ijms-24-00533-t001:** IC_50_ values from in vitro neuroprotective assays: anticholinesterase (AChE and BChE), anti-inflammatory (LOX), and antioxidant (ABTS, ROS, and RNS) activities.

	In Vitro Neuroprotection																	
Extract	AChE			BChE			LOX			ABTS			ROS			RNS		
	IC_50_ µg/mL																	
**PLE100**	137.1	±	8.1	147.0	±	7.5	76.11	±	10.4	13.54	±	0.8	4.38	±	0.4	1199.0	±	98.1
**OL-SS**	144.4	±	29.1	183.8	±	22.5	104.8	±	11.4	82.59	±	1.1	18.27	±	0.5	1036.0	±	114.2
**DS**	18.85	±	0.1	113.5	±	11.5	63.38	±	6.5	16.33	±	2.1	3.41	±	0.2	698.2	±	34.6
**ASFE**	4.23	±	0.1	1.20	±	0.1	4.37	±	0.3	0.11	±	0.0	1.56	±	0.1	3218.0	±	358.6
**T33**	97.46	±	6.8	85.46	±	2.7	48.30	±	1.7	6.33	±	0.0	2.54	±	0.1	599.0	±	5.9
**PPC1**	67.03	±	5.3	150.7	±	1.1	52.20	±	5.4	6.86	±	0.0	6.95	±	0.5	838.0	±	99.5
**Quercetin ***							125.7	±	10.8									
**Galantamine ***	0.40	±	0.0	2.36	±	0.0				4.56	±	0.4	0.98	±	0.1			
**Ascorbic acid ***																1100.9	±	13.9

*Citrus sinensis* waste (PLE100), *Olea europaea L.* leaves (OL-SS), *Dunaliella salina* (DS), cooperage woods of *Robinia pseudoacacia* (ASFE), *Cyphomandra betacea* peel (T33) and *Coffea arabica* silverskin (PPC1). * Control positive used for in vitro neuroprotective assays.

The large content of phenolic acids and flavonoids present in the ASFE and T33 extracts seems to be responsible for their notable neuroprotective capacities [12,43]. The presence of phenolic compounds, such as protocatechuic acid, ferulic acid and gallic acid, among others, was reported in brain tissues after oral administration experiments using rats [44,45], and caffeic acid and other flavonoids were identified in human cerebrospinal fluid (CSF) [38].

Several molecular parameters, such as TPSA, lipophilicity (calculated as log *p*), molecular weight (MW), pKa, hydrogen bond donors (HBD), hydrogen bond acceptors (HBA) and others factors have been suggested to affect the diffusion of the compounds across the BBB [46,47] (Table 2). Different authors have provided some “rules” to try to explain the accessibility of natural compounds to the BBB. For instance, Hitchcock suggested 90 Å^2^ as the limit value for TPSA in BBB permeability [48]; Agatonovic-Kustrin et al. highlighted that molecules with high Log *p* values could penetrate better through the BBB due to the lipophilic nature of this physiological barrier [42]; and Yang et al. described that the BBB transport values of the flavonoids are reduced when the numbers of glycoside and hydroxyl groups of the flavonoids increased [49]. These facts along with the high MW, TPSA and low log *p* could explain that rutin (in T33 extract), and hesperidin and kaempferol diacetylglucoside rhamnoside (in ASFE extract) have poor transport trough the BBB (Table 2). Moreover, some molecules, especially those with a high MW, tend to form more hydrogen bonds which restrict their diffusivity across a hydrophobic barrier such as the BBB [38] (as can be seen in Table 2 for rutin, hesperidin and kaempferol diacetylglucoside rhamnoside).

Nowadays, the mechanism by which phenolic compounds cross the BBB is not completely understood, and most authors suggest that the entry of phenolics compounds into the CNS cannot be explained by passive transport alone, but an active transport is requested. In this line, the interaction between flavonoids and some efflux transporters such as P-glycoprotein (P-gp) has been documented, and some reports have suggested that flavonoids act as substrates of these transporters (Table 2) [50]. Other interesting groups of compounds are tocopherols, which are predicted as not BBB permeable based on their TPSA and Log *p* values, but they can be P-gp substrates [51], and we observed that they cross the BBB (Table 2). However, this relationship requires additional investigations to be totally confirmed. Likewise, Figueira et al. reported that the presence of ATP binding cassette (ABC) efflux transporters in HBMEC play a key role in the transport of phenolic compounds across the BBB [27].

It should be noted that the above-mentioned “rules” are not absolute, and additional factors can modulate the compounds’ potential to cross the BBB. As discussed throughout this section, many natural compounds are capable of crossing the BBB, even though some of them do not fulfil all the required characteristics proposed. In this sense, the BBB permeability potential of many compounds could be affected by the compound concentration used in the experiments. Moreover, some competition mechanisms can appear, suggesting that one compound may alter the bioavailability of others [27,38].

On the other hand, the most valuable compounds with neuroprotective potential present in DS extract are carotenoids and their isomers [16,52], which were not detected in the lower compartment (“brain side”) (Figure 1b). Furthermore, we observed a decrease in the number of these compounds in the apical side, meaning that they could be adsorbed on the PAMPA-BBB membrane or the HBMEC monolayer. However, it is well established that carotenoids can be transported across the BBB into the CNS, as confirmed in post mortem human brain tissue analysis [53,54]. Carotenoids belong to a group of hydrophobic molecules with 40 carbon atoms that seem to need specific transporters to transit between the blood and the human tissues [55]. In this line, a carotenoids transporter still needs to be defined and several mechanisms seem to be involved in their cellular metabolism and transport, such as expression of the transcription factors that regulate transporter expression, or the presence of carrier proteins involved in carotenoid uptake [55]. All these conditions would require the use of a more complex BBB system closer to the in vivo models.

**Table 2 ijms-24-00533-t002:** Molecular parameters of the bioactive compounds detected and phytochemical profile of the analyzed extracts arranged by families and transport efficiency (Te) after 2 h and 4 h of incubation.

Compound (Family)	Matrix	Molecular Formula	Molecular Weight (*m*/*z*)	Log *p* *	TPSA * (Å^2^)	HBA *	HBD *	BBB Predicted ^a^	P-gp * Substrate ^b^	Te % (SEM)	
Terpenoids										2 h	4 h
**Monoterpenoids**											
3-Carene	PLE100	C_10_H_16_	136.1252	2.8	0	0	0	Yes	No	n.d	20.3 (0.9)
Limonene	PLE100	C_10_H_16_	136.1252	3.4	0	0	0	Yes	No	33.6 (2.3)	59.3 (2.9)
L-α-Terpineol	PLE100	C_10_H_18_O	154.1357	1.8	20.2	1	1	Yes	No	17.6 (1.3)	59.3 (0.4)
Nerol	PLE100	C_10_H_18_O	154.1357	2.9	20.2	1	1	Yes	No	26.8 (0.2)	41.5 (3.1)
Limonene Epoxide	PLE100	C_10_H_16_O	152.1201	2.5	12.5	1	0	Yes	No	19.5 (1.2)	42.4 (3.2)
Borneol	OL-SS	C_10_H_18_O	154.1357	2.7	20.2	1	1	Yes	No	8.5 (0.3)	35.2 (0.1)
Cymenol	OL-SS	C_10_H_14_O	150.1044	2.3	20.2	1	1	Yes	No	51.5 (1.8)	75.9 (2.9)
Thymol	OL-SS	C_10_H_14_O	150.1044	3.3	20.2	1	1	Yes	No	34.1 (0.1)	47.0 (1.7)
**Meroterpenoids**											
Chromene derivative	OL-SS	C_12_H_20_	164.1565	2.2	63.6					2.7 (0.0)	12.4 (0.8)
**Sesquiterpenoids**											
β-Panasinsene	PLE100	C_15_H_24_	204.1878	5.1	0	0	0	No	No	33.4 (1.5)	34.8 (1.0)
(-)-Aristolene	PLE100	C_15_H_24_	204.1878	4.7	0	0	0	No	No	39.7 (0.4)	71.3 (1.6)
Valencene	PLE100	C_15_H_24_	204.1878	5.2	0	0	0	No	No	51.9 (4.2)	86.1 (0.9)
(−)-α-Panasinsen	PLE100	C_15_H_24_	204.1878	4.9	0	0	0	No	No	19.8 (0.2)	83.5 (5.1)
γ-silenene	PLE100	C_15_H_24_	204.1878	5.4	0	0	0	No	No	52.8 (3.8)	53.8 (2.8)
Elemol	PLE100	C_15_H_26_O	222.1983	4.4	20.2	1	1	Yes	No	24.6 (0.5)	25.4 (1.6)
**Compound** **(Family)**	**Matrix**	**Molecular formula**	**Molecular weight (*m*/*z*)**	**Log *p* ***	**TPSA *** **(Å^2^)**	**HBA ***	**HBD ***	**BBB** **Predicted ^a^**	**P-gp * substrate ^b^**	**Te** **% (SEM)**	
δ-Cadinene	PLE100	C_15_H_24_	204.1878	5.5	0	0	0	No	No	21.5 (0.8)	28.9 (2.4)
β-Sinensal	PLE100	C_15_H_22_O	218.167	4.8	17.2	1	0	No	No	31.6 (1.2)	38.2 (2.4)
Nootkatone	PLE100	C_15_H_22_O	218.167	3.9	17.1	1	0	Yes	No	29.9 (0.9)	43.0 (3.4)
Ylangenal	PLE100	C_15_H_22_O	218.167	3.7	17.1	1	0	Yes	No	32.0 (2.7)	36.9 (0.4)
Nerolidol	OL-SS	C_15_H_26_O	222.1983	4.6	20.2	1	1	Yes	No	30.1 (2.5)	59.8 (2.4)
Farnesene	OL-SS	C_15_H_24_	204.1878	6.1	0	0	0	No	No	46.0 (2.1)	50.9 (3.1)
Caryophyllene oxide	OL-SS	C_15_H_24_O	220.1827	4.5	0	1	0	Yes	No	67.7 (0.9)	77.1 (3.9)
γ-Elemene	OL-SS	C_15_H_24_	204.1878	5.4	0	0	0	No	No	30.1 (0.1)	41.8 (1.7)
(-)-Globulol	OL-SS	C_15_H_26_O	222.1983	20.2	37.3	1	1	Yes	No	44.1 (3.6)	94.7 (3.5)
Hexahydrofarnesyl	OL-SS	C_18_H_36_O	268.2766	6.9	17.1	1	0	No	Yes	23.4 (1.1)	48.1 (3.0)
Germacrene D	OL-SS	C_15_H_24_	204.1878	4.7	0	0	0	No	No	3.2 (0.2)	6.6 (0.4)
Phytuberin	T33	C_17_H_26_O_4_	294.1831	2.3	45	4	0	Yes	No	33.4 (1.5)	79.6 (2.2)
**Diterpenoids**											
Geranylgeraniol	OL-SS	C_20_H_34_O	290.2609	6.6	20.2	1	1	Yes	No	30.2 (1.1)	56.6 (3.1)
Isophytol	OL-SS	C_20_H_40_O	296.3079	7.8	20.2	1	1	No	Yes	15.6 (1.0)	58.0 (1.7)
Phytol	OL-SS	C_20_H_40_O	296.3079	9.2	20.2	1	1	No	Yes	15.6 (0.3)	43.5 (3.7)
**Triterpenoids**											
Squalene	OL-SS/PLE100	C_30_H_50_	410.391	11.6	0	0	0	No	Yes	24.1 (1.4)/19.1 (1.4)	33.2 (0.9)/47.6 (1.5)
β-Amyrin	OL-SS/PLE100	C_30_H_50_O	426.3861	9.2	20.2	1	1	No	No	n.d	n.d
α-Amyrin	OL-SS	C_30_H_50_O	426.3861	9	21.2	1	1	No	No	n.d	n.d
Lupeol acetate	OL-SS	C_30_H_48_O_2_	440.3654	10.4	26.3	2	0	No	No	n.d	n.d
Uvaol Isomer I	OL-SS	C_30_H_50_O_2_	442.381	7.4	40.5	2	2	No	No	3.3 (0.1)	38.8 (3.2)
**Compound** **(Family)**	**Matrix**	**Molecular formula**	**Molecular weight (*m*/*z*)**	**Log *p* ***	**TPSA *** **(Å^2^)**	**HBA ***	**HBD ***	**BBB** **Predicted ^a^**	**P-gp * substrate ^b^**	**Te** **% (SEM)**	
Uvaol Isomer II	OL-SS	C_30_H_50_O_2_	442.381	7.4	40.5	2	2	No	No	17.7 (1.0)	38.3 (0.7)
Ganolucidic acid C	ASFE	C_30_H_46_O_7_	518.3243	3.1	132	7	4	No	Yes	11.7 (0.7)	15.6 (0.3)
Corosolic acid	ASFE	C_30_H_48_O_4_	472.3552	6.4	78	4	3	No	Yes	33.4 (1.9)	39.6 (1.9)
Triterpene ^#^	ASFE	C_29_H_44_O_5_	472.3188	3.6	84	5	2	No	Yes	30.4 (1.7)	82.9 (5.3)
Lucyin A	ASFE	C_30_H_46_O_5_	486.3345	5.3	95	5	3	No	Yes	7.6 (0.5)	25.3 (0.7)
Ganolucidic acid B	ASFE	C_30_H_46_O_6_	502.3294	3.8	112	6	3	No	Yes	28.4 (1.1)	28.6 (1.5)
**Tocopherols**											
β-Tocopherol	OL-SS	C_28_H_48_O_2_	416.3654	10.3	29.5	2	1	No	Yes	n.d	15.2 (0.8)
α-Tocopherol	OL-SS/PLE10	C_29_H_50_O_2_	430.381	10.7	29.5	2	1	No	Yes	19.8 (0.2)/18.3 (0.8)	39.7 (3.0)/29.1 (1.5)
γ-Tocopherol	OL-SS/PLE10	C_28_H_48_O_2_	416.1654	10.3	29.5	2	1	No	Yes	31.5 (0.8)/27.3 (2.1)	42.0 (0.8)/30.0 (1.6)
**Sesterterpenoids**											
Tocospiro A	OL-SS	C_29_H_50_O_4_	462.3709	7.4	63.6	4	1	No	Yes	2.7 (0.2)	25.6 (1.5)
Tocospiro B	OL-SS	C_29_H_50_O_4_	462.3709	7.4	63.6	4	1	No	Yes	1.9 (0.1)	26.2 (1.0)
**Apocarotenoid**											
Dihydroactinidiolide	OL-SS	C_11_H_16_O_2_	180.115	2.2	26.3	2	0	Yes	No	42.2 (3.5)	51.8 (3.0)
3-hydroxy-β-damascone	OL-SS	C_13_H_20_O_2_	208.1463	2	37.3	2	1	Yes	No	84.5 (3.2)	95.5 (3.2)
Isololiolide	OL-SS	C_11_H_16_O_3_	196.1099	1	46.5	3	1	Yes	No	33.0 (2.6)	61.4 (4.3)
**Phytosterols**											
Campesterol	PLE100	C_28_H_48_O	400.3705	8.8	20.2	1	1	No	No	68.7 (2.2)	76.7 (0.5)
Stigmasterol	PLE100	C_28_H_48_O	412.3705	8.6	20.2	1	1	No	No	40.4 (2.2)	64.5 (0.4)
γ-Sitosterol	OL-SS/PLE100	C_29_H_50_O	414.3861	9.3	20.2	1	1	No	No	12.9 (0.6)/11.3 (0.5)	19.2 (1.3)/12.1 (0.6)
**Compound** **(Family)**	**Matrix**	**Molecular formula**	**Molecular weight (*m*/*z*)**	**Log *p* ***	**TPSA *** **(Å^2^)**	**HBA ***	**HBD ***	**BBB** **Predicted ^a^**	**P-gp * substrate ^b^**	**Te** **% (SEM)**	
**Oxosteroids**											
Cucurbitacin	ASFE	C_30_H_42_O_6_	498.2981	3.2	101	8	3	No	Yes	41.1 (3.0)	75.4 (2.0)
**Phenolic compounds**											
**Hydroxycinnamic acids**											
p-Coumaric acid	T33	C_9_H_8_O_3_	164.0473	1.5	58	3	2	Yes	No	9.1 (0.6)	37.8 (2.5)
Caffeic acid	T33/ASFE	C_9_H_8_O_4_	180.0422	1.2	78	4	3	No	No	34.8 (1.2)/55.4 (3.7)	50.1 (3.4)/68.3(0.3)
Caffeoylshikimic acid	T33	C_16_H_16_O_8_	336.0845	0.9	147	8	4	No	No	14.3 (0.8)	29.0 (0.1)
Ethyl caffeate	T33	C_11_H_12_O_4_	208.0735	2.6	67	4	2	Yes	No	33.4 (2.4)	35.7 (0.3)
Caftaric acid	ASFE	C_13_H_12_O_9_	312.0481	0.1	162	9	5	No	No	46.3 (3.6)	87.6 (3.5)
Ferulic acid	ASFE	C_10_H_10_O_4_	194.0579	1.5	67	4	2	Yes	No	n.d	9.1 (0.3)
Coutaric acid	ASFE	C_13_H_12_O_8_	296.0532	0.4	141	8	4	No	No	n.d	n.d
Caffeoyl hexoside	T33	C_15_H_18_O_9_	342.0950	−0.6	157	9	6	No	No	67.4 (4.3)	72.8 (0.6)
**Hydroxybenzoic acids**											
p-Salycilic acid	T33	C_7_H_6_O_3_	138.0316	3.7	58	3	2	Yes	No	25.5 (1.8)	43.4 (1.3)
Sinapic acid	T33/ASFE	C_12_H_12_O_5_	236.0684	1.5	76	5	2	No	No	35.6 (2.0)/n.d	44.4 (3.2)/18.6 (0.5)
Protocatechuic aldehyde	ASFE	C_7_H_6_O_3_	138.0316	1.1	58	3	2	Yes	No	20.3 (1.5)	78.2 (5.7)
Protocatechuic acid	ASFE	C_7_H_6_O_4_	154.0266	1.1	78	4	3	No	No	45.1 (3.8)	71.7 (3.2)
**Methoxybenzoic acids**											
Gallic acid	T33/ASFE	C_7_H_6_O_5_	170.0215	0.7	98	5	4	No	No	33.0 (1.9)/39.9 (1.4)	62.9 (0.3)/67.9(1.3)
Vanillic acid	ASFE	C_8_H_8_O_4_	168.0422	1.4	67	4	2	No	No	38.9 (1.2)	42.2 (0.7)
**Compound** **(Family)**	**Matrix**	**Molecular formula**	**Molecular weight (*m*/*z*)**	**Log *p* ***	**TPSA *** **(Å^2^)**	**HBA ***	**HBD ***	**BBB** **Predicted ^a^**	**P-gp * substrate ^b^**	**Te** **% (SEM)**	
**Coumaric acids**											
Rosmarinic acid	T33	C_18_H_16_O_8_	360.0845	2.4	145	8	5	No	No	41.0 (2.3)	90.3 (4.2)
**Cyclic alcohols**											
Quinic acid	T33	C_7_H_12_O_6_	192.0633	−2.4	118	6	5	No	No	20.1 (1.6)	52.5 (1.4)
Caffeoylquinic acid	T33	C_16_H_18_O_9_	354.0950	−0.4	165	9	6	No	No	15.7 (0.4)	74.8 (2.9)
O-acetyl-quinic acid	T33	C_9_H_14_O_7_	234.0739	−2.4	124	7	4	No	Yes	15.6 (1.2)	23.7 (0.7)
Methyl caffeoyl quinate	T33	C_17_H_20_O_9_	368.1107	1.9	200	9	5	No	No	5.2 (0.2)	12.5 (0.2)
Methyl feruloyl quinate	T33	C_18_H_22_O_9_	382.1263	0.2	143	9	4	No	Yes	26.5 (1.7)	76.5 (5.0)
**Methoxyphenols**											
Syringaldehyde	T33/ASFE	C_9_H_10_O_4_	182.0579	0.1	56	4	1	Yes	No	22.7 (1.5)/33.4 (2.2)	45.7 (3.2)/42.0 (1.4)
**Flavonoid O-glycosides**											
Rutin	T33	C_27_H_30_O_16_	610.1533	−1.3	266	16	10	No	Yes	n.d	n.d
Isoquercitrin	T33	C_21_H_20_O_12_	464.0954	0.4	207	12	8	No	No	6.0 (0.2)	16.3 (1.3)
Kaempferol rhamnoside	ASFE	C_31_H_34_O_17_	678.1796	0.1	245	17	7	No	Yes	n.d	n.d
Hesperidin	ASFE	C_28_H_34_O_15_	610.1897	−1.1	234	15	8	No	Yes	n.d	n.d
**O-methylated flavonoids**											
Hesperetin	ASFE	C_16_H_14_O_6_	302.0790	2.4	96	6	3	No	Yes	65.5 (4.8)	67.2 (3.6)
**Flavonols**											
Kaempferol	ASFE	C_15_H_10_O_6_	286.0477	1.9	107	6	4	No	No	31.5 (1.1)	36.5 (2.2)
**Flavanonols**											
**Compound** **(Family)**	**Matrix**	**Molecular formula**	**Molecular weight (*m*/*z*)**	**Log *p* ***	**TPSA *** **(Å^2^)**	**HBA ***	**HBD ***	**BBB** **Predicted ^a^**	**P-gp * substrate ^b^**	**Te** **% (SEM)**	
Aromadendrin	ASFE	C_15_H_12_O_6_	288.0633	1.8	107	6	4	No	No	76.2 (0.8)	78.7 (0.7)
**Biflavonoids**											
Procyanidin B2	ASFE	C_30_H_26_O_12_	578.1424	2.4	221	12	10	No	No	56.8 (0.0)	61.9 (2.6)
**Hydrolyzable tannins**											
Syringic acid hexoside	T33	C_15_H_20_O_10_	360.1056	−1	155	10	5	No	No	92.0 (0.7)	94.9 (3.1)
Ellagic acid	ASFE	C_14_H_6_O_8_	302.0062	1.1	134	8	4	No	No	42.4 (1.6)	85.5 (2.5)
**Benzenetriols**											
Pyrogallol	ASFE	C_6_H_6_O_3_	126.0316	0.5	61	3	3	Yes	No	37.6 (2.6)	80.5 (1.1)
**Chalcones**											
Butein	ASFE	C_15_H_12_O_5_	272.0684	2.8	98	5	4	No	No	27.5 (2.0)	36.3 (2.6)
**Stilbenes**											
e-viniferin	ASFE	C_28_H_22_O_6_	454.1416	5.4	110	6	5	No	No	39.2 (2.2)	45.9 (3.0)

# (23S,24S)-17,23-Epoxy-24,29-dihydroxy-27-norlanost-8-ene-3,15-dione). * Log *p* (octanol–water partition coefficient). * TPSA (topological polar surface area). * HBA (hydrogen bond acceptors). * HBD (hydrogen bond donors). * P-gp (P-glycoprotein transporters). ^a,b^ Computational models for permeability BBB and P-gp substrate prediction [51,56]. Cooperage woods of Robinia pseudoacacia (ASFE), Olea europaea L. leaves (OL-SS), Cyphomandra betacea peel (T33) and Citrus sinensis waste (PLE100).

## 3. Materials and Methods

### 3.1. Natural Biomass and Extraction Conditions

The extracts from T33, PPC1, and PLE100 tested along this work were prepared trough PLE extraction in a Dionex accelerated solvent extractor equipped with a stainless-steel extraction cell (ASE 200, Sunnyvale, CA, USA). Briefly, the dried and grinded residues were mixed with sea sand (1:2 *w*/*w*) and placed into the extraction cell. All the extractions were obtained in static mode at constant pressure (10 MPa) using ethyl acetate at 100 °C for 30 min in the case of PLE100; and pure ethanol (EtOH) at 180 °C and 150 °C for 20 min in the case of T33 and PPC1 extracts, respectively. Finally, 1 min of nitrogen purging was used to push any residual solvent from exhausted powder residue.

DS, ASFE and OL-SS extracts were obtained by SFE processes. In short, CO_2_ was pumped into the extraction cell by a high-pressure pump until reaching the desired values of pressure and temperature. In the case of ASFE, the temperature was 60 °C in the extraction cell, the extraction pressure was 23.5 MPa, and 45% of co-solvent [EtOH/H_2_O (1:1, *v*/*v*)] was used for 60 min of extraction time. With regard to DS extract, the pressure was 40 MPa at 45 °C for 2 h. In the case of OL-SS, the extractions conditions were 30 MPa and 60 °C for 2 h. Finally, all the extracts were collected in glass vials and stored at—20 °C in the dark until drying by N_2_. The optimum extraction conditions were selected according to previous experiments described by our group to achieve the maximum in vitro neuroprotective capacity for all these biomasses [12,13,14,15,52]. Obtained dried extracts were diluted in EtOH at a final concentration of 0.2% (*v*/*v*).

### 3.2. Chemicals and Reagents

Roswell Park Memorial Institute (RPMI) 1640 medium, fetal bovine serum (FBS), non-essential amino acids (NEAA), minimal essential medium (MEM) vitamins, sodium pyruvate, _L_-glutamine, trypsin-EDTA, antibiotic-antimycotic solution, formic acid, NaCl, KCl, CaCl_2_·H_2_O, MgCl_2_·6H_2_O, D-glucose, Hepes, NaHCO_3_, thiazolyl blue tetrazolium bromide (MTT) and sodium fluorescein (molecular weight, 376 Da) were purchased from Sigma-Aldrich (St. Louis, MO, USA). NuSerum IV was obtained from Corning Costar Corp (New York, NY, USA). HCl was purchased from Fisher Chemical (Waltham, MA, USA). HPLC-grade solvents EtOH, isopropanol, methyl tert-butyl ether (MTBE), methanol and ethyl acetate were acquired from VWR Chemicals (Barcelona, Spain). Ultrapure water was obtained from a Millipore system (Billerica, MA, USA). For SFE, premier quality CO_2_ was provided by Carburos Metálicos (Madrid, Spain). Sea sand of SFE and PLE extraction was obtained by Panreac Quimica (Barcelona, Spain). Ultrapure water was obtained from a Millipore system (Billerica, MA, USA). For SFE, premier quality CO_2_ was provided by Carburos Metaálicos (Madrid, Spain). Sea sand of SFE and PLE extractions was obtained by Panreac Quimica (Barcelona, Spain).

### 3.3. Cell Culture Conditions

The cell line HBMEC was used as a simplified in vitro model of the BBB endothelium. This cell line was immortalized with SV40 large T antigen [57]. Cells were cultured in RPMI 1640 medium supplemented with 10% (*v*/*v*) FBS, 10% (*v*/*v*) NuSerum IV, 1% (*v*/*v*) NEAA, 1% (*v*/*v*) MEM, 1 mM sodium pyruvate, 2 mM l-glutamine and 1% (*v*/*v*) antibiotic-antimycotic solution. Endothelial cell cultures were maintained at 37 °C in a humid atmosphere enriched with 5% CO_2_. In our studies, HBMEC were used between passage 25 and 29. For experiments, HBMECs were seeded using a volume of 200 µL at a density of 8 × 10^4^ cells/mL in rat tail collagen-I (100 µg/mL) coated 96-well plates (for viability assay) or 500 µL at a density of 1.6 × 10^5^ cells/mL in rat tail collagen-I (100 µg/mL) coated semi-permeable membranes in transwell inserts (0.4 µm pore, Corning Costar Corp., USA; for BBB integrity and transport assays) for 12-well plates. The incubation timepoints were assay dependent.

### 3.4. Viability Assay

The effect of each extract on HBMEC viability was evaluated via an MTT assay. Forty-eight hours after seeding, cells were incubated with the extracts at different concentrations, diluted in RPMI, or with RPMI (with 0.2% ethanol; control), and then incubated for 4 h at 37 °C and 5% CO_2_. Then, the media were removed and RPMI containing 0.5 mg/mL of MTT was added to each well and incubated for 3 h at 37 °C and 5% CO_2_. The supernatant was removed, and the formed formazan crystals in viable cells were solubilized with 0.04 N HCl in isopropanol solution. Finally, absorbance values at 595 nm were measured using a microplate reader (Varioskan LUX Multimode Microplate reader, ThermoFisher Scientific, Waltham, MA, USA). The values of cell viability were calculated as a percentage relative to the control (RPMI with EtOH at not exceeding 0.2%, *v*/*v*). The data was averaged from three independent experiments, performed in triplicate.

### 3.5. Blood–Brain Barrier Transport Studies

For the transendothelial transport assays, HBMEC were seeded on transwell inserts as described above and are schematically depicted in Figure 1a. After seeding, the media were changed every two days. Four days post-seeding, the HBMEC monolayer was formed, and the transport studies were performed. In brief, the media were removed and 0.5 mL of extracts at a selected concentration diluted in RPMI, or RPMI only (control), were added to the upper compartment and the lower compartment was filled with 1.5 mL of RPMI, and then incubated for 2 and 4 h. At the end of each timepoint, the media from both compartments were collected and lyophilized in a freeze-drier for 24 h at −84.5 °C and 4800 Pa (Lyobeta 15 Telstar, Terrassa, Spain), and the samples were then stored at −80 °C until analysis. The bioactive compound transport efficiency (Te) across the HBMEC monolayer was determined by Equation (1), adapted from Pogačnik et al. [28]:(1)$$\mathrm{Te}=\frac{\frac{C_{lower}}{V_{upper}}}{\frac{C_{upper}}{V_{lower}}}\times 100\;\%$$
where *C_lower_* and *C_upper_* is the relative concentration (peak area) of the compound in lower and upper compartments, respectively. *V_upper_* and *V_lower_* is the upper and lower compartment volume (0.5 and 1.5 mL, respectively). The culture inserts were used for the subsequent integrity permeability studies.

### 3.6. Blood–Brain Barrier Integrity

The transendothelial electrical resistance (TEER) and sodium fluorescein (Na-F) paracellular permeability, two recognized BBB integrity parameters [27,28], were assessed to evaluate the effect of each extract on BBB properties and to discard false positives regarding the presence in the lower compartment of any natural compound due to disruption of the HBMEC monolayer.

#### 3.6.1. Transendothelial Electrical Resistance

TEER measurements were performed as previously described [27]. The electrical resistance across the endothelial monolayer was assessed using an EndOhm™ chamber coupled to a resistance meter (EVOMX; World Precision Instruments, Inc, Sarasota, FL, USA). The TEER readings were collected before (0 h) and after (2 and 4 h) extracts addition. After subtracting the values of empty inserts (without cells and extracts) and multiplying by the area of the insert (1.12 cm^2^), TEER values expressed as Ω × cm^2^ were obtained and the percentage of variation from average control readings (without extracts) were determined. After the last reading, the media from both compartments were collected and used for the transport studies mentioned above, and then the empty inserts were used for the following permeability assay.

#### 3.6.2. Sodium Fluorescein Paracellular Permeability

The Na-F permeability assay described by Veszelka et al. can be used as a marker of the paracellular permeability [30]. Briefly, after transport and TEER assays described in Section 3.5 and Section 3.6.1, the media were collected and the inserts were washed twice with Ringer–Hepes solution (150 mM NaCl, 5.2 mM KCl, 2.2 mM CaCl_2_.H_2_O, 0.2 mM MgCl_2_·6H_2_O, 2.8 mM D-glucose, 5 mM Hepes, 6 mM NaHCO_3_ at pH 7.4) in both compartments. Then, the inserts were carefully placed in new 12-well plates containing 1.5 mL of Ringer–Hepes solution, and 10 µg/mL of Na-F in Ringer–Hepes solution was added to the upper compartments (0.5 mL) and incubated at 37 °C, protected from light. The inserts were transferred to new wells containing Ringer–Hepes solution at 20, 40 and 60 min. After these times, the inserts were removed and the solutions from the lower compartment were collected to determine fluorescein levels using a fluorescence spectrophotometer (Varioskan LUX Multimode Microplate Reader, ThermoFisher Scientific, USA; excitation: 460 nm and emission: 515 nm). The Na-F flux across the untreated HBMEC monolayers (inserts without extract) and the cell-free inserts were also measured. The transendothelial permeability coefficient (Pe) was calculated as previously described [58]. In short, the transport of Na-F was expressed as microliter of tracer diffusing from the upper to lower chambers (Equation (2)).
Clearance (µL) = [Concentration_Lower_] × [Volume_Lower_] × [Concentration_upper_]^−1^(2)

Then, average cleared volume was plotted versus time, and permeability × surface area product value for the endothelial monolayer (PSe) was calculated by Equation (3):PS_endothelial_^−1^ = PS_time − plotted−1_ − PS_insert_^−1^(3)

Thereafter, PS_e_ was divided by the insert surface area (1.12 cm^2^) to give the endothelial permeability coefficient (Pe in 10^−6^ cm/s), which was expressed as a percentage of variation from the control (inserts without extract).

### 3.7. Quantification of Bioactive Compounds in BBB Transport Assays

#### 3.7.1. Bioactive Compound Extraction

With the aim of recovering and concentrating the bioactive compounds present in the upper and lower compartments, as well as avoiding possible interferences from the RMPI medium in the analysis, different extraction methods were applied depending on the nature of the extract. For extracts enriched in low polar bioactive compounds, such as terpenoids from PLE100 and OL-SS and carotenoids from DS, extraction using 50:50 ethyl acetate/H_2_O was employed. In more detail, mass freeze-dried samples placed in 2 mL Eppendorf tubes were reconstituted by the addition of 0.5 mL of cold H_2_O (vortex 10 s) and 0.5 mL of cold ethyl acetate (vortex 10 s). Thereafter, samples were centrifuged for 5 min at 14,000 rpm. After centrifugation, the supernatant (ethyl acetate) was transferred to a 1.5 mL Eppendorf tube and dried in a SpeedVac system at 40 °C and 13 mbar (SPD1030, Thermo Fisher Scientific). Finally, the supernatant enriched in low polar bioactive compounds was resuspended in 25 µL of pure ethanol prior to the analysis. On the other hand, for extracts enriched in more polar compounds, such as phenolic compounds from T33, ASFE and PPC1, extraction using 80:20 methanol/H_2_O was employed. The extraction process is similar to the one described above, except for the use of methanol instead of ethyl acetate.

#### 3.7.2. Gas Chromatography–Mass Spectrometry (GC–MS)

Based on the phytochemical profiling of terpenoids from the target orange and olive leave by-products reported in our previous works [14,15], the content of terpenoids in the samples from the upper and lower compartments from PLE100 and OL-SS treatment was determined. Samples were dissolved in pure ethanol and analyzed in an Agilent 7890B GC system coupled to an Agilent 7200 q-TOF MS, equipped with an electronic impact (EI) ionization source. An Agilent Zorbax DB5- MS + 10 m Duraguard Capillary Column (30 m × 250 μm × 0.25 μm) was used for terpenoid separation. The carrier gas was helium at constant flow rate of 0.8 mL min^−1^. The injector operated in splitless mode for 2 min with the injector temperature at 250 °C. The injection volume was 1 μL. The GC oven was set at 60 °C for 1 min, then the temperature was increased at a rate of 10 °C min^−1^ until 325 °C, and this temperature was maintained for 10 min. The MS analyzer was operated in full-scan acquisition mode at a *m*/*z* scan range of 50–600 Da (5 spectra per second). The temperatures of the quadrupole, the transfer line and the ion source were established at 150, 290 and 250 °C, respectively. Each extract was injected in duplicate. Quantitative analysis was achieved using the Agilent Mass Hunter Quantitative analysis software for Q-TOF (version B.08.00).

#### 3.7.3. Liquid Chromatography Tandem Mass Spectrometry (UHPLC-q-TOF-MS)

The analysis of T33, ASFE, PPC1 and DS extracts was performed by UHPLC (Agilent 1290 UHPLC system) coupled to a q-TOF-MS/MS analyzer (Agilent 6540 q-TOF MS) using conditions similar to those previously described [12,13,16]. For PPC1, T33 and ASFE extracts, 5 µL (in duplicate) was injected and separated in a Zorbax Eclipse Plus C18 column (2.1 × 100 mm, 1.8 m particle diameter, Agilent Technologies, Santa Clara, CA, USA). Separation was carried out at 30 °C at a constant flow rate of 0.5 mL min^−1^. The mobile phase was composed of water (0.01% formic acid) as solvent A and acetonitrile (in 0.01% formic acid) as solvent B. The gradient elution program was as follows: 0 min, 0% B; 7 min, 30% B; 9 min, 80% B; 11 min, 100% B; 13 min, 100% B; 14 min, 0% B. An orthogonal electrospray ionization (ESI) source (from Agilent Technologies) was used for MS analysis with the following parameters: 10 mL min^−1^ drying in gas flow rate; 4000 V as the capillary voltage; the nebulizer pressure was 40 psi; 350 °C was the gas temperature; the skimmer voltage was 45 V; and the fragmentor voltage was 110 V. The MS mode was used at a scan rate of 5 spectra per second and *m*/*z* values were acquired ranging between 50 and 1100.

In the case of DS extract, 10 μL (in duplicate) was injected and separated in a YMC-C30 reversed-phase analytical column (250 mm × 4.6 mm i.d., 5 μm particle size; YMC Europe, Schermbeck, Germany) with a YMC-C30 (10 mm × 4 mm i.d., 5 μm particle size) pre-column. A solution of methanol/MTBE/water (90:7:3 *v*/*v*/*v*) was used as mobile phase A, while methanol/MTBE (10:90 *v*/*v*) was employed as mobile phase B. The flow rate was 0.8 mL min^−1^ and the temperature was held at 30 °C. The chromatographic separation was run according to the following gradient: 0 min, 30% B; 15 min, 40% B; 18 min, 80% B; 23 min, 100% B; 25 min, 0% B; 28 min, 30% B; and 30 min, 30% B. An orthogonal atmospheric-pressure chemical ionization (APCI) source (from Agilent Technologies) was used for MS analysis with the following parameters: 8 L/min drying in gas flow rate; 3500 V as the capillary voltage; the nebulizer pressure was 40 psi; 300 °C as the gas temperature; the skimmer voltage was 45 V; and the fragmentor voltage was 110 V. The MS mode was used at a scan rate of 3 spectra per second and *m*/*z* values were acquired ranging between 50 and 1500. After MS acquisition, data were processed by Agilent Mass Hunter mentioned above in Section 3.7.2. Quantitative Analysis software was used to obtain relative abundances for the tentatively identified bioactive compounds.

Additional molecular information about each natural compound, such as topological polar surface area (TPSA) or Log *p* (octanol–water partition coefficient), was noted from the PubChem database https://pubchem.ncbi.nlm.nih.gov/ (accessed on 11 November 2022) and SwissADME database [56].

### 3.8. Statistical Analysis

Results were analyzed using GraphPad Prism^®^ 6.0 (GraphPad Software, San Diego, CA, USA) and are given as mean ± SEM (standard error of the mean). The results represent the average of at least three independent experiments, with three technical replicates in the case of the viability assay. One- and two-way ANOVA analyses were conducted to compare the mean values by Dunnett’s test. Statistically significant differences were considered when *p* < 0.05.

## 4. Conclusions

In the present work, different extracts obtained by the green extraction techniques from five agrifood by-products and one type of microalgae were evaluated through a cellular model of BBB to investigate the brain transport potential of their natural bioactive constituents. Our results demonstrate a reduction of Na-F permeability of HBMEC cell monolayers treated with PLE100 and T33 extracts, which might be derived from their high phytosterol and phenolic content. On the other hand, PPC1 was the only extract that damaged the HBMEC monolayer by the increment of Na-F permeability, which might be caused by its high caffeine content. Furthermore, a broad range of phytochemicals, such as phenolic acids, flavonols and low MW terpenoids, were detected in the “brain side” of the HBMEC-BBB model after 4 h, suggesting their high potential to reach the brain and apply their neuroprotection properties. Finally, some physicochemical properties of the bioactive molecules were related to their BBB permeability, such as the high MW, the high TPSA, the low log *p* values, or the presence of excessive HBD and HBA. The obtained results show the importance of natural matrixes as sources of bioactive compounds, with the ability to cross the BBB and reach the brain tissues to exert the claimed neuroprotective capacity. This research on the target natural extracts and their bioactive constituents provides further evidence of their neuroprotective properties due to their demonstrated potential to reach the CNS. However, further in vivo research is needed to explore the transport and molecular neuroprotective mechanisms of these compounds as promising sources of new functional foods and nutraceuticals with health-promoting properties against neurodegenerative disorders such as AD.

## Figures and Tables

**Figure 1 ijms-24-00533-f001:**
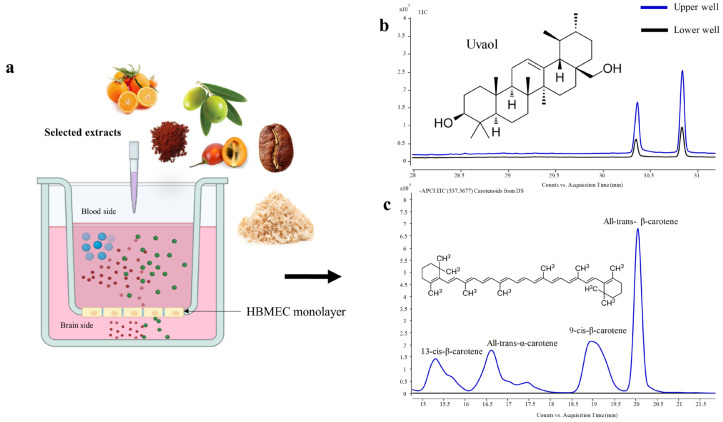
Schematic diagram of the experimental conditions and profiles obtained. The cell line of HBMEC was plated onto a semi-permeable membrane (transwell insert) placed inside a cell culture well defining two chambers: an upper or luminal one mimicking the blood compartment, and a lower or basolateral one that mimics the brain compartment (**a**). Uvaol from Olea europaea L. leaves (OL-SS) (total ion chromatogram, TIC) and carotenoids from Dunaliella salina (DS) (extracted ion chromatogram, EIC 547.3677 MW) were added to the upper chamber and their content was determined in the upper and lower chambers ((**b**) and (**c**), respectively).

**Figure 2 ijms-24-00533-f002:**
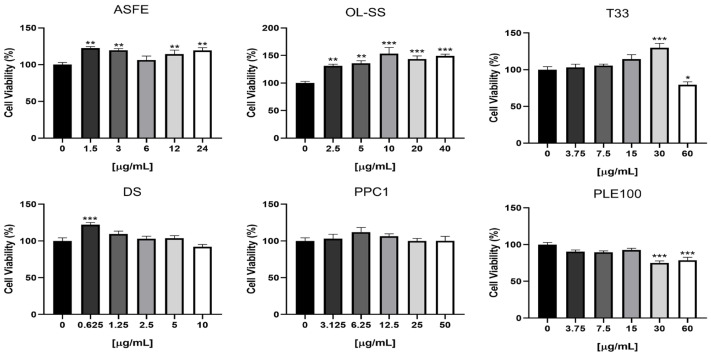
HBMEC viability upon exposure to the extracts at different concentrations. HBMEC cells were treated with the indicated concentrations of each extract: cooperage woods of Robinia pseudoacacia (ASFE), Olea europaea L. leaves (OL-SS), Cyphomandra betacea peel (T33), Dunaliella salina (DS), Coffea arabica silverskin (PPC1), and Citrus sinensis waste (PLE100), diluted in RPMI, or RPMI only (control), for 4 h. Cell viability was assessed by the MTT assay, and the values are presented as a percentage relative to untreated cells (control). All values are given as mean ± SEM of three independent experiments performed in triplicate. Statistical analysis was performed by one-way ANOVA. Significance is shown as * *p* < 0.05, ** *p* < 0.01 and *** *p* < 0.001 vs. control.

**Figure 3 ijms-24-00533-f003:**
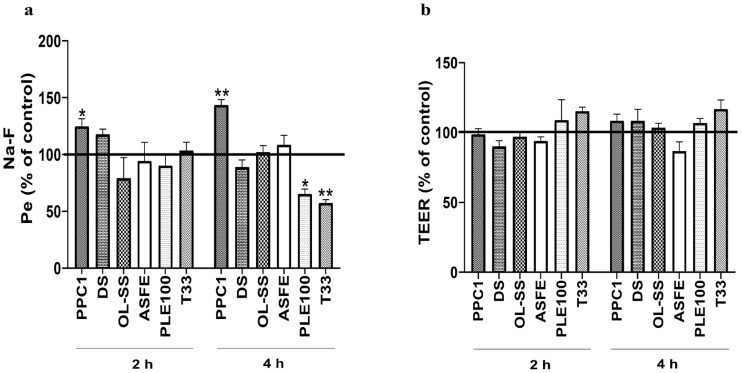
Effect of extracts in blood–brain barrier (BBB) integrity. Confluent HBMEC monolayers seeded onto semi-permeable membranes were treated with the indicated concentrations of each extract: Coffea arabica silverskin (PPC1; 50 µg/mL), Dunaliella salina (DS; 10 µg/mL), Olea europaea L. leaves (OL-SS; 40 µg/mL), cooperage woods of Robinia pseudoacacia (ASFE; 24 µg/mL), Citrus sinensis waste (PLE100; 15 µg/mL) and Cyphomandra betacea peel (T33; 30 µg/mL) diluted in RPMI, or RPMI only (control), for 2 and 4 h. Alterations in BBB properties were evaluated by endothelial paracellular permeability to sodium fluorescein (Na-F) (**a**) and trasendothelial electrical resistance (TEER) measurements (**b**). Quantitative analysis of both parameters revealed that PPC1 is the only extract that impairs the HBMEC endothelium, whereas PLE100 and T33 improve the BBB restricted permeability. Data are expressed as mean ± SEM of three independent experiments. Statistical analysis was performed by two-way ANOVA and the significance is shown as * *p* < 0.05, ** *p* < 0.01 vs. control.

**Figure 4 ijms-24-00533-f004:**
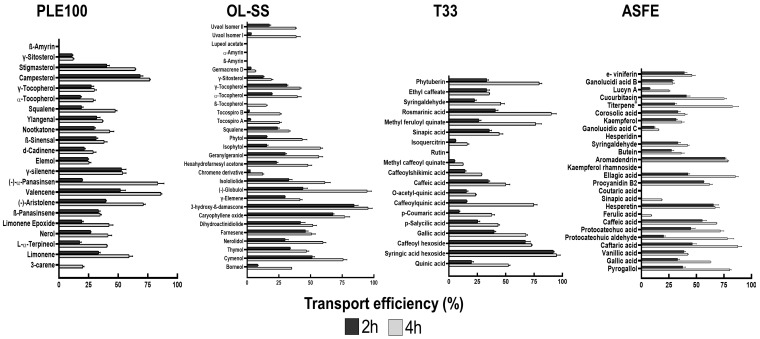
Bioactive compound transport efficiency (Te) from the different studied extracts, across the blood–brain barrier (BBB) endothelium. Confluent HBMEC monolayers seeded onto semi-permeable membranes were treated with the indicated concentrations of each extract: Citrus sinensis waste (PLE100; 15 µg/mL), Olea europaea L. leaves (OL-SS; 40 µg/mL), Cyphomandra betacea peel (T33; 30 µg/mL), and cooperage woods of Robinia pseudoacacia (ASFE; 24 µg/mL) diluted in RPMI for 2 and 4 h. The upper and lower compartment solutions were collected, analyzed by GC/LC-Q-TOF-MS and a quantitative analysis of the concentration (peak area) of each compound between the lower (“brain side”) and upper (“blood-side”) compartments was performed.

## Data Availability

Not applicable.

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
