# Peer review of "In Vitro Study of the Blood–Brain Barrier Transport of Natural Compounds Recovered from Agrifood By-Products and Microalgae"

_ijms, 2022, doi:10.3390/ijms24010533_

Round 1

Reviewer 1 Report

Comments and suggestions

This article is well written. Some modifications are needed for improvement.

Abstract 

Abstract should be exact in wording, and must be understandable to a wide audience.

1. Line 26, e blood-brain barrier (BBB) you already abbreviated line 22.

2. In vitro italic is necessary? Please check and confirm.

Introduction

Please write the specific aim of your study end of the introduction. Also, carefully checked all the pants and compounds name has written correctly.

Results and discussion

1. Figure 1(C)  is not clear. Please provide a good resolution figure.

2. 2.1. Evaluation of agrifood by-products and microalgae extracts toxicity on HBMEC. Why different concentrations used? Significance is shown as *p<0.05, **p<0.01 and ***p<0.001 vs. control. Please check p value has written correctly.

3. Figure 3  is not clear. Please provide a good resolution figure. 

Materials and Methods

Please carefully check and confirm all chemicals and reagents name has been written correctly.

Conclusion should be more concise. Please demonstrate the importance of your study. 

Author Response

The authors appreciate the valuable comments and suggestions from the reviewers, which have greatly contributed to improve the quality of the manuscript. The manuscript was thoroughly revised and appropriate corrections were made. All changes done in the revised manuscript have been marked using the "Track Changes" tool from Word. The comments are answered below.

REVIEWER 1

Abstract 

Abstract should be exact in wording, and must be understandable to a wide audience.

The authors appreciate this comment and abstract was rewriting to make it more understandable to a wider audience.

  1. Line 26, e blood-brain barrier (BBB) you already abbreviated line 22.

The authors appreciate this observation, and redundant information was removed.

  1. In vitro italic is necessary? Please check and confirm.

Yes, the italic font is commonly used all along the scientific literature for the Latinism in vitro.  Also for in vivo and in silico experiments.

Introduction

Please write the specific aim of your study end of the introductionAlso, carefully checked all the pants and compounds name has written correctly.

The aim of this manuscript is pointed out in lines 111 -115:

“The aim of the present work is to evaluate the BBB permeability of different natural compounds from agrifood by-products and microalgae extracts by using the HBMEC cell line model in a Transwell-based system. Preliminary studies about the potential cellular cy-totoxicity and the effect on the BBB integrity (barrier tightness and paracellular permeabil-ity) were also investigated”

Results and discussion

  1. Figure 1(C)  is not clear. Please provide a good resolution figure.

Figure 1(C) was resized to improve quality. All figures are submitted to the journal in .TIFF format at the highest resolution to guaranty the maximum quality in the final version.

2.1. Evaluation of agrifood by-products and microalgae extracts toxicity on HBMEC. Why different concentrations used? Significance is shown as *p<0.05, **p<0.01 and ***p<0.001 vs. control. Please check p value has written correctly.

We chose different extract concentrations in toxicity assays based on the different cytotoxicity potential of each extract due to their inherent composition. The selected initial concentration tested in viability assays was based on previous studies with other cell lines, as pointed out in line 129: “The tested concentrations were selected based on previous viability tests performed for the same extracts on other human cell lines, namely neuron-like (SH-SY5Y), kidney (HK-2) and monocyte THP-1) cells [12–16] “

Regarding the p value, this point was carefully revised and different p values were given to highlight different levels of statistical significance (* for p<0.05, ** for p<0.01 and *** for p<0.001).

  1. Figure 3  is not clear. Please provide a good resolution figure. 

In line with the comment above, figure 3 was resized to improve quality. In any case, all figures are submitted to the journal in .TIFF format at the highest resolution to guaranty the maximum quality in the final version.

Materials and Methods

Please carefully check and confirm all chemicals and reagents name has been written correctly.

As suggested, chemicals and reagents 3.2 section was carefully revised and all the chemical names were double-checked.

Conclusion should be more concise. Please demonstrate the importance of your study. 

The authors appreciate this observation, and conclusion section was revised.

Reviewer 2 Report

The work is interesting, well structured and methodologically adequate. The limitation, however admitted by the same authors, concerns the validation of these results in vivo, which can certify the efficacy of the compounds as well as the possibility of crossing the bbb. One thing they could add in the discussion and in the conclusion is whether these compounds can be proposed as a supplement for a therapeutic or preventive effect against inflammatory neurodegenerative diseases.

Author Response

The authors appreciate the valuable comments and suggestions from the reviewers, which have greatly contributed to improve the quality of the manuscript. The manuscript was thoroughly revised and appropriate corrections were made. All changes done in the revised manuscript have been marked using the "Track Changes" tool from Word. The comments are answered below.

REVIEWER 2

The work is interesting, well structured and methodologically adequate. The limitation, however admitted by the same authors, concerns the validation of these results in vivo, which can certify the efficacy of the compounds as well as the possibility of crossing the bbb. One thing they could add in the discussion and in the conclusion is whether these compounds can be proposed as a supplement for a therapeutic or preventive effect against inflammatory neurodegenerative diseases.

The authors very much appreciate this valuable suggestion. The high value-added bioactive compounds present in the studied extracts were shown to be promising functional ingredients that might be deliver into the marked as food supplements, food additives, or nutraceuticals with scientifically supported neuroprotective claims. This information has been added to the conclusion section.

Reviewer 3 Report

The title of this paper is "In vitro study of the blood-brain barrier transport of natural compounds recovered from agrifood by-products and microalgae". ASFE, OL-SS, T33, DS, PPC1, and PLE100 extracts were studied in an in vitro BBB cell model to investigate the brain transport potential of bioactive substances. The novelty of the paper was acknowledged and well written.

1. The manufacturing process of ASFE, OL-SS, T33, DS, PPC1, and PLE100 extracts is described in detail in Materials and methods.

2. In the introduction, you should also add the reasons for choosing ASFE, OL-SS, T33, DS, PPC1 and PLE100.

3. Neuronal protection results against phytochemicals derived from ASFE, OL-SS, T33, DS, PPC1 and PLE100 extracts should be added to the discussion section. If there are possible test results, they should be presented together.

Author Response

The authors appreciate the valuable comments and suggestions from the reviewers, which have greatly contributed to improve the quality of the manuscript. The manuscript was thoroughly revised and appropriate corrections were made. All changes done in the revised manuscript have been marked using the "Track Changes" tool from Word. The comments are answered below.

REVIEWER 3

  1. The manufacturing process of ASFE, OL-SS, T33, DS, PPC1, and PLE100 extracts is described in detail in Materials and methods.

The authors appreciate this comment and more information about the extraction process of the extracts was added in section 3.1.

  1. In the introduction, you should also add the reasons for choosing ASFE, OL-SS, T33, DS, PPC1 and PLE100.

This information appears in the introduction section between lines 71 and 80:

Previous studies by our group revealed that terpenoids, carotenoinds and phenolic compounds from agrifood by-products, such as unroasted cooperage woods of Robinia pseudoacacia (ASFE), Cyphomandra betacea peel (T33), Coffea arabica silverskin (PPC1), Olea europaea L. leaves (OL-SS), Citrus sinensis wasted (PLE100) and the microalgae Dunaliella salina (DS) show in vitro neuroprotective potential [12–16]. These extracts showed a promising cholinesterase and LOX inhibition capacity, and high antioxidant activity by scavenging reactive oxygen and nitrogen species (ROS and RNS, respectively). Also, OL-SS, PLE100 and DS extracts have demonstrated neuroprotective activity in human neuroblastoma SH-SY5Y cell lines against Aβ1–42 and L-glutamic acid associated toxicity [15–17]

  1. Neuronal protection results against phytochemicals derived from ASFE, OL-SS, T33, DS, PPC1 and PLE100 extracts should be added to the discussion section. If there are possible test results, they should be presented together.

The authors appreciate the valuable comment from the reviewer, which have greatly contributed to improve the quality of the manuscript. According to this comment, a new table summarizing in vitro neuroprotective results of the target extracts was added to the discussion. The new table 1 in the manuscript shows IC50 values from in anticholinesterase (AChE and BChE), anti-inflammatory (LOX), and antioxidant (ABTS, ROS, and RNS) assays.